# Metabolomic and Transcriptomic Analyses Reveal the Characteristics of Tea Flavonoids and Caffeine Accumulation and Regulation between Chinese Varieties (*Camellia sinensis* var. *sinensis*) and *Assam* Varieties (*C. sinensis* var. *assamica*)

**DOI:** 10.3390/genes13111994

**Published:** 2022-10-31

**Authors:** Hao Tang, Man Zhang, Jiayu Liu, Jiao Cai

**Affiliations:** Guangdong Provincial Key Laboratory of Tea Plant Resources Innovation and Utilization, Tea Research Institute, Guangdong Academy of Agricultural Sciences, Guangzhou 510640, China

**Keywords:** *Camellia sinensis*, flavonoids, catechin, caffeine, *TCS1*

## Abstract

Flavonoids and caffeine are the major secondary metabolites with beneficial bioactivity for human health in tea plants, and their biosynthesis pathway and regulatory networks have been well-deciphered. However, the accumulation traits of flavonoids and caffeine in different tea cultivars was insufficient in investigation. In this study, metabolomic and transcriptomic analyses were performed to investigate the differences of flavonoids and caffeine accumulation and regulation between Chinese varieties, including the ‘BTSC’ group with green leaf, the ‘BTZY’ group with purple foliage, and the ‘MYC’ group comprising *Assam* varieties with green leaf. The results showed that most of the flavonoids were down-regulated in the ‘MYC’ group; however, the total anthocyanin contents were higher than that of the ‘BTSC’ group while lower than that of the ‘BTZY’ group. An *ANS* (*Anthocyanin synthase*) was significantly up-regulated and supposed to play a key role for anthocyanin accumulation in the ‘BTZY’ group. In addition, the results showed that esterified catechins were accumulated in the ‘BTSC’ and ‘BTZY’ groups with high abundance. In addition, *SCPL1A* (*Type 1A serine carboxypeptidase-like acyltransferases* gene) and *UGGT* (*UDP glucose: galloyl-1-O-β-d-glucosyltransferase* gene) potentially contributed to the up-accumulation of catechins esterified by gallic acid. Interestingly, the results found that much lower levels of caffeine accumulation were observed in the ‘MYC’ group. RT-qPCR analysis suggested that the expression deficiency of *TCS1* (*Tea caffeine synthase 1*) was the key factor resulting in the insufficient accumulation of caffeine in the ‘MYC’ group. Multiple MYB/MYB-like elements were discovered in the promoter region of *TCS1* and most of the MYB genes were found preferentially expressed in ‘MYC’ groups, indicating some of which potentially served as negative factor(s) for biosynthesis of caffeine in tea plants. The present study uncovers the characteristics of metabolite accumulation and the key regulatory network, which provide a research reference to the selection and breeding of tea varieties.

## 1. Introduction

Tea is the oldest and most popular nonalcoholic and caffeine-containing beverage and is consumed daily by more than 3 billion people around the world [1]. Many of the unique secondary metabolites in tea leaves, such as polyphenols, caffeine, theanine, vitamins, glycosides, and minerals, contribute to its attractive aroma and healthful benefits [2,3,4]. Among these metabolites, theanine, catechins, and caffeine are three major characteristic components in tea. Theanine is the most abundant non-proteinogenic amino acid in tea and offers the characteristic umami flavor contributing to the bitter and astringent taste [5]. Catechins, belonging to the polyphenols, account for 12 to 24% of dried young leaves, and most of them are esterified by gallic acid [6]. Catechins in tea are mixtures of various esterified forms. The most abundant catechins are generally (−)-epigallocatchin-3-gallate (EGCG) and (−)-epicatechin-3-gallate (ECG), followed by (−)-epigallocatechin (EGC), (+)-gallocatechin (GC), (+)-catechin (C) and (−)-epicatechin (EC) [7]. As a main antioxidant agent, catechins remove harmful reactive oxygen components in the human body and play the beneficial activity for prevention of neurodegeneration disease [8] and inhibition of cancer proliferation [9]. In black tea, catechins are enzymatically oxidized to o-quinone and further polymerized to form theaflavin and thearubigins, which are characterized as quality components with function of blood-lipid reduction and anti-inflammatory properties in black tea [10]. In addition, catechins have been proposed to improve the tolerance of tea plants under drought stress by directly scaveninge excess ROS during polymerization and formation of metal ion complexes [11]. Caffeine is the predominantly accumulated purine alkaloid in young shoots of tea plants and their related species, while its content varies within the range of 6.2–27.7 mg/g of dried leaf [12]. Caffeine usually offers a part of bitter taste of tea beverage and is usually used as a class of stimulant for improvement of the central nervous system excitation [6]. However, for some people, excessive caffeine intake may interfere with sleep and even cause caffeine poisoning (caffeinism), which includes symptoms such as anxiety, agitation, insomnia, and gastrointestinal disorders [13,14]. Therefore, caffeine is an explicit breeding target in order to obtain new varieties with low caffeine content.

Based on the genome assembly of *C. sinensis* and comparative study among other *Camellia* species, higher expression levels of many flavonoids, catechins, and caffeine biosynthesis-related genes contributing to the quality of tea have been well-elucidated [6,15]. Genetic factors, development stages, and different abiotic stressors significantly affect accumulation of flavonoids and catechins in tea plants and thereby influence the collection and processing of various tea types. Molecular studies have shown that the flavonoids and catechins are derived from the phenylpropanoid pathway, in which they share common early enzymatic genes, such as chalcone synthase (*CHS*), chalcone isomerase (*CHI*), and dihydroflavonol reductase (*DFR*) [6,16]. The later genes, leucoanthocyanidin reductase (*LAR*) and anthocyanidin reductase (*ANR*), are contributed to catechins biosynthesis [6]. Meanwhile, enzymatic genes are invariably mastered by the MYB-bHLH-WD40 (MBW) complex [17,18]. Other transcription factors (TFs), such as WRKY [19] and NAC [20], have also been supposed to participate in pathway regulating flavonoids and catechins biosynthesis. The caffeine biosynthesis pathway has been well-elucidated, in which xanthosine, *S*-adenosylmethionine serving as methyl donor, is orderly methylated at the *N*-7, *N*-3, and *N*-1 position. As the first *N*-methyltransferase gene, TCS1 has been supposed to be the key enzyme which catalyzes the methylation of *N*-3 and *N*-1 [21]. Furthermore, an MYB family gene (*CsMYB184*) has been proved to regulate caffeine biosynthesis in tea plants by the function of *TCS1* promoter activation [22]. To meet the personalized needs of different groups, breeding of the new germplasm with abundant catechins and low caffeine has been one of the goals for genetic improvement. Therefore, a profound deciphering of the molecular mechanism for secondary metabolite biosynthesis is highly expected.

Here, in order to investigate the metabolite accumulation differences and regulatory networks among various tea plants, we collected the typical small-leaf cultivars belonging to the Chinese varieties and large-leaf cultivar belonging to *Assam* varieties. Widely targeted metabolomics technology was first used to investigate the contents of flavonoids, and results showed that most of the metabolites were down-regulated in *Assam* varieties and the content of anthocyanins was expected to largely accumulate in Chinese varieties with purple young leaf. We also found that *SCPL1A* and *UGGT* potentially commonly contributed to positively regulate the galloylated catechins accumulation in Chinese varieties based on transcriptomic analyses. Interestingly, we found that the caffeine was barely accumulated due to the expression deficiency of *TCS1* in *Assam* varieties. Multiple MYB family genes were preferentially expressed in *Assam* varieties, implying that they potentially served as a negative regulatory factor of caffeine biosynthesis. This research provides available information for improving the beneficial metabolite accumulation in tea plants.

## 2. Materials and Methods

### 2.1. Plant Materials

Nine tea cultivars were used as materials to perform widely targeted metabolomic analyses, transcriptome sequencing, and determination of secondary metabolites content. All local cultivars were planted in the Huizhou region [Guangdong, China (114°30′–114°53′ E, 22°26′–22°32′ N)]. All cultivars tested in this study were divided into three groups according to their own characteristics, including ‘Baitangshancha (BTSC)’, ‘Baitangziyacha (BTZY)’, and ‘Maoyecha (MYC)’. Of which, ‘BTSC’ contains ‘BTSC-1’, ‘BTSC-2’, and ‘BTSC-3’ cultivars (*C. sinensis* var. *sinensis*). ‘BTZY’ contains ‘BTZY-1’, ‘BTZY-2’, and ‘BTZY-3’ cultivars (*C. sinensis* var. *sinensis*). ‘MYC’ contains ‘MYC-1’, ‘MYC-2’, and ‘MYC-3’ cultivars (*C. sinensis* var. *assamica*). All materials were grown under the natural environmental conditions and typical phenotypes were described in Results. Young buds with two leaves of every cultivar were sampled and frequently frozen in liquid nitrogen. All samples were stored at −80 °C until metabolite extraction, metabolomic analyses, and transcriptomic analyses.

### 2.2. Metabolite Extraction and MS Detection

Metabolites extraction was according to previous methods and modified [23]. Freeze-dried leaves were smashed, and 100 mg of the powder was weighed and extracted overnight at 4 °C with 1.0 mL of 70% aqueous methanol. After being centrifuged at 10,000× *g* for 10 min, the extracts were absorbed on a CNWBOND Carbon-GCB SPE Cartridge (250 mg, 3 mL; ANPEL, Shanghai, China) and filtered (SCAA-104, 0.22 μm pore size; ANPEL, Shanghai, China) for LC-MS analysis. The LC-ESI-MS/MS system (HPLC, Shim-pack UFLC SHIMADZU CBM30A system, www.shimadzu.com.cn/ (accessed on 5 May 2020); MS, Applied Biosystems 4500 Q TRAP, http://www.appliedbiosystems.com.cn/ (accessed on 5 May 2020)) was used to analyzed sample extracts. The liquid chromatography analytical conditions were based on the previous description [24].

Linear ion trap (LIT) and triple quadrupole (QQQ) scans were acquired on a triple quadrupole-linear ion trap mass spectrometer (API 4500 Q TRAP LC/MS/MS System; Boston, MA, USA) equipped with an ESI Turbo Ion Spray interface, operating in both positive and negative ion modes and controlled by Analyst 1.6.3 software (AB Sciex, Singapore). The ESI source operation parameters were as follows: ion source, turbo spray; source temperature, 550 °C; ion spray voltage (IS), 5500 V; and ion source gas I (GSI), gas II (GSII), curtain gas (CUR) set to 55, 60 and 25 psi, respectively. The collision activated dissociation (CAD) was set at “high”. Instrument tuning and mass calibration were performed with 10 and 100 μmol/L polypropylene glycol solutions in QQQ and LIT modes, respectively. QQQ scans were acquired via multiple reaction monitoring (MRM) experiments with collision gas (nitrogen) set to 5 psi. Declustering potential (DP) and collision energy (CE) were optimized for individual MRM transitions. A specific set of MRM transitions was monitored for each period according to the metabolites eluted within the period [23].

### 2.3. MS Data and Statistical Analyses

MS data acquisition and processing were conducted as described previously [25]. The analyses of the primary and secondary MS data were performed based on the self-built database MWDB (Metware Biotechnology Co., Ltd. Wuhan, China). Metabolite quantification was accomplished with data acquired in MRM mode by QQQ-MS [26]. Metabolites with a fold change ≥2 or a fold change ≤0.5 were identified as upregulated or downregulated [23].

### 2.4. Library Preparation for RNA Sequencing

The RNA sequencing (RNA-seq) libraries were constructed using an NEBNext^®^ UltraTM RNA Library Prep Kit for Illumina^®^ (NEB, Ipswich, MA, USA) following the manufacturer’s descriptions. Briefly, messenger RNA (mRNA) was purified from total RNA using poly-T oligo attached magnetic beads. Fragmentation was performed using divalent cations under elevated temperature in NEBNext First Strand Synthesis Reaction Buffer (5×). First strand cDNA was synthesized using random hexamer primer and M-MuLV Reverse Transcriptase (RNase H−). Second strand cDNA synthesis was subsequently performed using DNA Polymerase I and the RNA was digested by RNase H. Remaining overhangs were converted into blunt ends. After adenylation of 3′ ends of DNA fragments, NEBNext Adaptor with hairpin loop structure were ligated to prepare for hybridization. In order to select cDNA fragments of preferentially 250~300 bp in length, the library fragments were purified with AMPure XP system (Beckman Coulter, Shanghai, China). Then 3 µL USER Enzyme (NEB, USA) was used with size-selected, adaptor-ligated cDNA at 37 °C for 15 min followed by 5 min at 95 °C before PCR. Then PCR was performed with Phusion High-Fidelity DNA polymerase, Universal PCR primers and Index (X) Primer. At last, PCR products were purified by AMPure XP system (Beckman Coulter, Shanghai, China) and library quality was assessed on the Agilent Bioanalyzer 2100 system. The library preparations were sequenced on an Illumina^®^HiSeq2500 platform and 125 bp/150 bp paired-end reads were generated.

### 2.5. Function Annotation and Expression Analysis

The raw data were filtered by removing low-quality reads and adaptors and were changed into clean reads. the HISAT v2.1.0 package was used to construct the index and map clean reads to the NCBI GeneBank. The feature Counts v1.6.2 package was used to count the gene alignment and then calculate the Fragments Per Kilobase per Million (FPKM) of each gene based on the gene length. The DESeq2 v1.22.1 was used to analyze the differential expression between the two groups, and the *p*-value was corrected using the Benjamini and Hochberg method. Gene function was annotated according to these databases: NCBI non-redundant protein sequences (Nr); Clusters of Orthologous Groups of proteins (COG/KOG); Swiss PROT protein sequence database (Swissprot); Kyoto Encyclopedia of Genes and Genomes (KEGG); homologous protein family (Pfam) and Gene Ontology (GO).

### 2.6. Measurements of Other Main Metabolites

The content of catechins, caffeine, and amino acids were determined by the HPLC system based on the previous reported methods [27]. The content of total anthocyanins was determined by the pH-based method [28]. Simply, 1 g sample was sliced and extracted with 15 mL Extract Buffer (0.15% HCl: 95%methanol, *v*:*v* = 15:85) for 4 h. The mixture was centrifuged, and its absorbance was determined at 530, 620, and 650 nm, respectively. The anthocyanin content was calculated based on the formula: ∆A/mL = (A_530_ − A_620_) − 0.1 (A_650_ − A_620_).

## 3. Results

### 3.1. Comparison of Morphological Phenotype and Metabolomic Analysis

To effectively develop the local tea plant resources, nine local tea plant varieties were collected and evaluated, including three ‘BTSC’ varieties (BTSC-1, BTSC-2, BTSC-3), three ‘BTZY’ varieties (BTZY-1, BTZY-2, BTZY-3), and three ‘MYC’ varieties (MYC-1, MYC-2, MYC-3). Significant phenotypic differences were observed among the three groups, especially focusing on the color of buds and leaves. In ‘BTSC’ groups, ‘BTSC-3’ and ‘BTSC-1’ showed green young leaves and rosiness in young stems, and the ‘BTSC-2’ exhibited green young leaves stems (Figure 1A). In the ‘BTZY’ group, all varieties had purple buds and young leaves; however, the mature leaves on the lower part were green (Figure 1A). For the ‘MYC’ group, their buds, young leaves, and stems were also green (Figure 1A).

In tea plants, purple leaves are usually due to anthocyanin accumulation. Therefore, metabolic profiling was first conducted to analyze flavonoids accumulation and focus on anthocyanin contents among the three groups. A total of 259 metabolites were identified from nine samples based on the UPLC-ESI-MS/MS system (Appendix A). According to the metabolome data, metabolites were divided into the following categories: 97 flavonols, 72 flavones, 22 flavone carbonosides, 16 flavanols, 15 proanthocyanidins, 13 anthocyanins, 7 dihydroflavones, 7 tannins, 4 dihydroflavonols, 3 isoflavones, 2 chalcones, and 1 anthraquinone (Appendix A). To obviously detect the metabolites accumulation pattern among the three groups, hierarchical cluster analysis was conducted based on the logarithm of the peak area matrix for each metabolite. As shown in Figure 1B, a high degree of consistency of metabolite accumulation pattern was observed among groups. The flavonoids accumulation pattern of the ‘MYC’ group was obviously distinct from the ‘BTSC’ and ‘BTZY’ group; most of the flavonoids were down-regulated (Figure 1B).

### 3.2. Differential Accumulated Metabolites Analyses

To identify the differentially accumulated metabolites (DAMs), OPLS-DA (orthogonal projection to latent structures-discriminant analysis) was used to perform between sample groups. the value of Q^2^ for all groups (BTSC_vs._BTZY, BTSC_vs._MYC, BTZY_vs._MYC) were more than 0.8 (Appendix A). Based on this algorithm, all groups were clearly separated and placed within a confidence interval, indicating that DAMs can be filtrated based on the variable importance in projection (VIP) obtained in OPLS-DA mode (Appendix A). A fold change ≥2 (up-regulated) or ≤0.5 (down-regulated) were served as standard, and these DAMs were filtered by the VIP value (VIP ≥ 1) from the OPLS-DA model. The results of DAMs for each sample group were presented in heatmap (Appendix A). As results, 55 significantly DAMs were identified in the ‘BTZY’ group compared to the ‘BTSC’ group (29 up-regulated, 26 down-regulated) (Appendix A). 105 DAMs were found in the ‘BTSC’ group compared to the ‘MYC’ group, however, only 8 DAMs were up-accumulated in ‘MYC’ group (Appendix A). 110 DAMs were found in the ‘BTZY’ group compared to the ‘MYC’ group, and most DAMs (103 DAMs) were also down-accumulated in ‘MYC’ group (Appendix A). The Kyoto Encyclopedia of Genes and Genomes (KEGG) metabolic pathway analyses indicated that DAMs were mainly enriched into the flavonoids synthesis pathway, including isoflavonoid biosynthesis, flavonoid biosynthesis, flavone and flavonol biosynthesis, and anthocyanin biosynthesis (Appendix A). Those results suggested that the flavonoids accumulation in the ‘MYC’ group was typically of lower abundance compared with the ‘BTSC’ and ‘BTZY’ group.

Anthocyanins are the primary pigments that have been proved to determine the coloration of tea plants, therefore, which are typically focused. Six main types of anthocyanidin aglycone, namely cyanidin (Cy), delphinidin (Dp), pelargonidin (Pg), peonidin (Pn), petunidin (Pt), and malvidin (Mv), were found (Figure 2). The differentially accumulated anthocyanin was logically abundant in the ‘BTZY’ group with purple-leaf, including cyanidin-3-*O*-glucoside, cyanidin-*O*-syringicacid, cyanidin-3-*O*-rutinoside, cyanidin-3-*O*-caffeoylsophoroside, malvidin-3-*O*-glucoside, pelargonidin-3-*O*-(6′-acetylglucoside), and pelargonidin-3, 5-diglucoside (Figure 2A–C,F,G,L,M). Compared with ‘BTSC’ group, cyanidin-3-*O*-rutinoside, pelargonidin-3-O-(6′-acetylglucoside), and pelargonidin-3, 5-diglucoside were scarcely accumulated in ‘MYC’ group (Figure 2C,L,M). As well as cyanidin-3-*O*-glucoside, cyanidin-*O*-syringicacid, delphinidin-3-*O*-glucoside, and pelargonidin-3-*O*-glucoside were also significantly up-accumulated in ‘MYC’ group comparing with ‘BTSC’ group (Figure 2A,B,I,K).

### 3.3. Measurement of Bioactive Components

In order to comprehensively explore the accumulation characteristics of metabolites in these tea plants, the content of metabolites, including catechin, anthocyanin, polyphenol, amino acid, and caffeine, were detected (Figure 3). Results indicated that the content of total catechin exhibited no difference among the three groups (Figure 3A), but the contents of EGC, EGCG, and ECG in the ‘BTSC’ and ‘BTZY’ groups were significantly higher than in the ‘MYC’ group (Figure 3C,E,G). The content of L-catechin (L-C) was dramatically up-accumulated in the ‘MYC’ group (Figure 3D). Compared with the ‘BTSC’ and ‘MYC’ group, the content of anthocyanin was observably much more abundant in ‘BTZY’ (Figure 3H). Additionally, the contents of water extract and polyphenols were down-regulated in the ‘MYC’ group (Figure 3I,J). Notably, only a small amount of caffeine accumulation was found in the ‘MYC’ group compared with the ‘BTSC’ and ‘BTZY’ groups (Figure 3L). Those results indicated that the galacylated catechins were abundant in the ‘BTSC’ and ‘BTZY’ groups and, in particular, the caffeine content was lower in the ‘MYC’ group.

### 3.4. Identification of Differentially Expressed Genes

For RNA-seq among three groups, high-quality total RNA of young leaves was reverse-transcribed into cDNAs and further amplified to construct 9 cDNA libraries. A total of 44.5–71.2 million raw reads were obtained using the Illumina^®^HiSeq2500 platform, with an average value of 68.14 million each sample (Appendix A). After processing raw reads, 6.60–10.5 G clean base with a Q30 percentage of 90.04–93.22%. The GC content percentage of 44.58–45.18% were obtained and been available for analyses (Appendix A). All the reads were successfully mapped to the reference genome with percentage of 85.90–89.46%, and 76.11–83.79% reads were uniquely mapped (Appendix A). These results showed that high-quality sequencing data can be used for further analyses.

### 3.5. Functional Annotation of Differentially Expressed Genes

Totally, 50, 525 genes were identified by mapping to the reference genome of ‘Shuchazao’ (*C. sinensis* var. sinensis) (Appendix A). The differentially expressed genes (DEGs) among the sample groups were defined by the fold-change of the FPKM value. In BTSC_vs._BTZY, 5935 DEGs were found, of which 2702 were up-regulated, and 3233 were down-regulated. In BTSC_vs._MYC, 9520 DEGs were detected, of which 4591 were up-regulated, and 4929 were down-regulated. In BTZY_vs._MYC, 9700 DEGs were noted, of which 5107 were up-regulated, and 4593 were down-regulated (Figure 4A). Furthermore, a total of 1012 DEGs were commonly identified in all comparable groups (Figure 4B).

The obtained unigenes were further used to constructed GO and KEGG analyses against the annotated canonical pathways. KEGG analyses suggested that 9, 11, and 13 metabolic pathways were significantly enriched in the BTSC_vs._BTZY, BTSC_vs._MYC, and BTZY_vs._MYC groups, respectively (Figure 4D–F). These pathways included phenylpropanoid biosynthesis, starch and sucrose metabolism, cyanoamino acid metabolism, zeatin biosynthesis, and ABC transporters in BTSC_vs._BTZY (Figure 4C). In BTSC_vs._MYC and BTZY_vs._MYC groups, similar pathways were enriched, including phenylpropanoid biosynthesis, carbon fixation, glutathione metabolism, amino acid metabolism and tyrosine metabolism (Figure 4E,F). Notably, A substantial number of DEGs were enriched into phenylpropanoid biosynthesis, which probably lead to the accumulation differences of anthocyanin and catechin.

GO is generally used for gene functional classification, include biological process (BP), molecular function (MF), and cellular component (CC). In BP, 2, 11, and 2 GO terms were maximally enriched in the BTSC_vs._BTZY, BTSC_vs._MYC, and BTZY_vs._MYC groups, respectively (Appendix A). In MF, iron ion binding and antiporter activity were the common GO terms maximally enriched (Appendix A). In CC, the GO term, apoplast, was significantly enriched in BTSC_vs._MYC and BTZY_vs._MYC groups (Appendix A).

### 3.6. Expression of the Genes on Anthocyanin and Catechin Biosynthesis

In order to investigate what genes contributed to the accumulation difference of anthocyanin and catechin among the three groups, the synthetase genes involved in anthocyanin and catechin biosynthesis were identified (Appendix A). The expression patterns of genes involved in biosynthesis pathway of anthocyanin were first analyzed among the three groups. According to the results, five anthocyanin-related synthetase genes with higher FPKM values were differentially expressed. The *CHS* (CSS0017197) in the early pathway was highly expressed in the ‘BTSC’ and ‘BTZY’ groups (Figure 5A,B). Two *F3′H* genes (CSS0030176 and CSS0048905) were highly expressed in ‘MYC’ groups with higher catechin (L-C) (Figure 3D and Figure 5A,B). One *DFR* (CSS0000672) and two *OMT* (CSS0027035 and CSS0033028) were also up-regulated in groups with higher total anthocyanin content (Figure 5A,B). Notably, an anthocyanin synthase gene (*ANS*, CSS0010687), which has been proved to be a key gene responsible for anthocyanin biosynthesis in tea plants, was significantly up-regulated in ‘BTZY’ comparing with the ‘BTSC’ group with lower anthocyanin content (Figure 5A,B). These results implied that the differential expression of the *ANS* results in abundant accumulation of anthocyanin in ‘BTZY’ group.

Catechins were valuable secondary metabolites in tea plants, and greatly conferred to the astringent taste and medicinal value in tea. The expression patterns of the key genes, *LAR* and *ANR* responsible for the biosynthesis of (+)-catechin (C), (−)-epicatechin (EC), (+)-gallocatechin (GC), and (−)-epigallocatechin (EGC), were first investigated (Appendix A and Figure 5C). Our results showed that two *LAR* genes (CSS0013831 and CSS0034690) were significantly up-regulated in ‘MYC’ group (Figure 5A,C), which is consistent with the abundant accumulation of (+)-catechin in ‘MYC’ group (Figure 3D). It was found that four *ANR* genes were up-regulated in both ‘BTSC’ and ‘BTZY’ group, potential resulting in the up-accumulation of EGC (Figure 3C and Figure 5A,C). For galloylated catechins, although the expression of five *SCPL1A* genes (CSS0041940, CSS0041635, CSS0032817, CSS0020267 and CSS0010497) were significantly up-regulated in ‘MYC’ groups (Figure 5A,C), however, the content of ECG and EGCG was lower abundance (Figure 3E,G). It was putative that these genes were probably irrelevant galloylated modification of catechin. In addition, the expression patterns of two *SCPL1A* genes (CSS0042199 and CSS0049169) were consistent with the accumulation of galloylated catechins. Interestingly, four *UGGT* genes (CSS0004941, CSS0009705, CSS0023751 and CSS0031621) were significantly up-regulated in ‘BTSC’ and ‘BTZY’ groups (Figure 5A), which potentially served as the key factors for up-regulation of galloylated catechins.

### 3.7. Regulation of Caffeine Biosynthesis in ‘MYC’ Group

The caffeine biosynthetic pathway has been well-investigated in tea plants. In the study, the caffeine content was 3.78–4.15% (*w*/*w*) in the ‘BTSC’ group and was 3.58–3.61% in the ‘BTZY’ group. However, much lower levels (0.05–0.55%) of caffeine accumulation were found in the ‘MYC’ group, which has caught our attention. To further investigate the reason for the accumulation lack of caffeine in ‘MYC’ tea plants, we identified 23 putative S-adenosylmethionine-dependent methyltransferase genes (Appendix A). However, the expression levels of all genes were not significantly variable among groups (Appendix A). Furthermore, caffeine synthase genes were considered to be the key factor responsible for caffeine biosynthesis by catalyzing the methylation reaction of theobromine. Regretfully, only a caffeine synthase gene (CSS0032602) (Appendix A) without expression pattern differences was identified in this study, which probably results from the incomplete assembling and annotation of reference genome. A previous study suggested that *TCS1* (accession number: XP_028084199.1) was the predominant key gene for caffeine biosynthesis, we then examine the expression pattern by RT–qPCR. The result found that the *TCS1* was barely expressed in ‘MYC’ group comparing with ‘BTSC’ and ‘BTZY’ group (Figure 6A). It was reasonably putative that the expression deficiency of *TCS1* contributed to the remarkable down-accumulation of caffeine (Figure 3L). The promoter elements of *TCS1* were also analyzed and found that multiple responsive elements were annotated in the upstream 2 kb (Figure 6B). These elements probably respond to different biological processes, including ABA responsiveness, light responsiveness, MeJA (jasmonic acid methyl ester) responsiveness, defensive and stress responsiveness, auxin responsiveness, and zein metabolism (Figure 6B). Interestingly, six MYB/MYB–like elements were found in the promoter region, therefore, the expression patterns of MYB family genes identified in the study were examined. The results showed that multiple genes were preferentially expressed in ‘MYC’ group, among them, seven genes were abundant accumulation, including CSS0040991, CSS0049966, CSS0014730, CSS0041456, CSS0013880, CSS0014476, and CSS0028845 (Figure 6C and Appendix A). The results implied that these MYB genes potentially served as negative factors involved in the biosynthesis of caffeine by inhibiting the expression of *TCS1*. Furthermore, three MYB family genes (CSS0028845, CSS0026012, and CSS0008558) were up-regulated in both ‘BTSC’ and ‘BTZY’ groups compared to the ‘MYC’ group (Figure 6C and Appendix A). Those results suggested that the expression deficiency of *TCS1* was the key factor for catalyzing the methylation of theobromine and the MYB family genes probably served as a significant regulatory factor.

## 4. Discussion

The purple leaf color of tea plants is mediated by specific genes’ expression, resulting in anthocyanin accumulation [29]. Integrative analysis of metabolome and transcriptome profiles has been widely performed to decipher the mechanism of color formation in plants [16,29,30]. In the present study, we first collected different endemic tea germplasms, including the ‘BTSC’ group, ‘BTZY’ group, and ‘MYC’ group. The morphological phenotypes, typically the color differences (Figure 1A), among these groups have been our focus. Therefore, metabolomic analysis was performed to investigate flavonoid accumulation differences. Results suggested that most of the flavonoids were significantly down-accumulated in the ‘MYC’ group (Figure 1B). Furthermore, 13 anthocyanins identified in the present research were abundantly accumulated in the ‘BTZY’ group with purple leaf. Notably, petunidin 3-*O*-glucoside, which was less identified in purple leaf tea plants, was also identified. This result consistent with ‘Zikui’ purple tea cultivar [31]. Furthermore, most of anthocyanins were down-regulated in the ‘BTSC’ group (Figure 2) and the total anthocyanin content was also lower than that of the ‘BTZY’ group (Figure 3H). Those results suggested that the anthocyanin accumulation differences contributed to the purple leaf color in the ‘BTZY’ group. In the previous study, anthocyanin hyperaccumulation was found in ‘Zijuan’ cultivars, which is one of the typical purple tea cultivars derived from a mutant of the *Assam* variety with large leaves in the Yunnan province of China [32]. However, in this study, the ‘BTZY’ group with purple leaves belongs to a Chinese variety. Interestingly, although the young leaves of the ‘MYC’ group was green, while some anthocyanins (cyanidin-3-*O*-glucoside, cyanidin-*O*-syringicacid, delphinidin-3-*O*-glucoside, and pelargonidin-3-*O*-glucoside) were also up-regulated compared to the ‘BTSC’ group (Figure 2) and the total anthocyanin content was lower than that of the ‘BTZY’ group (Figure 3H). The expression patterns of *F3′H* and *DFR* were also up-regulated in the ‘MYC’ group comparired with the ‘BTSC’ and ‘BTZY’ group. The key gene *ANS* (CSS0010687) was the highest expression level in the ‘BTZY’ group (Appendix A), which is probably responsible for the anthocyanin accumulation differences between the ‘BTZY’ and ‘MYC’ group.

Catechins are important metabolites in determining the flavor and health benefits of tea, and the biosynthesis mechanism in tea plants have been well-investigated [6]. In this study, the content of total catechin was evaluated and there was no difference among the three groups (Figure 3A). Furthermore, the content L-C, EC, ECG, EGC, and EGCG were also detected, respectively (Figure 3). We found that the content of L-C was significantly accumulated in the ‘MYC’ group, which was probably owed to the up-regulation of multiple genes (*F3′H*, *DFR*, and *LAR*) in the ‘MYC’ group (Figure 5A,B) [6]. The content of other esterified catechins, including ECG and EGCG as well as non-esterified EGC, was much higher in ‘BTSC’ and ‘BTZY’ than that of the ‘MYC’ group (Figure 3). Previous studies suggested that an enzyme, belonging to subclade 1A of serine carboxypeptidase-like acyltransferases (SCPL), was shown to play a key role in flavan-3-ol galloylation [33,34]. Galloylated catechins were biosynthesized via 1-*O*-glucose ester-dependent two-step reactions by acyltransferases, which involved UGGT and ECGT [35]. In this study, two *SCPL1A* genes were identified and potentially involved in the accumulation of the galloylated catechins. Furthermore, four *UGGT* genes, which catalyzed the biosynthesis of the galloylated acyl donor *β*-glucogallin, were also up-regulated in the ‘BTSC’ and ‘BTZY’ groups. Therefore, the higher expression level of *SCPL1A* and *UGGT* genes potentially contributed to the abundant accumulation of galloylated catechins in the ‘BTSC’ and ‘BTZY’ groups than that of the ‘MYC’ group.

Caffeine is an important flavor compound in tea, and the recent studies suggested that its content was less affected by environments; rather it was influenced by genotype [36]. In the present study, only a very small amount of caffeine was found to accumulate in the ‘MYC’ group (Figure 3L). The pathway of caffeine biosynthesis and metabolism were extensively investigated in the tea plant and its related species [21,27]. The typical biosynthetic pathways of caffeine are from xanthosine to caffeine by successive methylation reaction [37]. *N*-methyltransferases (NMTs) were proved to catalyze the methylation reaction in this pathway with the methyl donor of *S*-adenosyl-*L*-methionine [27]. In this study, putative *S*-adenosylmethionine-dependent methyltransferase genes were identified and no significant expression pattern association with the accumulation level was found among all cultivars (Appendix A). Based on the previous studies, the caffeine synthase gene *TCS* was proved to play a critical role for converting theobromine to caffeine [38]. However, only one annotated *TCS2* gene (CSS0032602) without expression pattern difference among all cultivars was found (Appendix A). Previous studies suggested that the biosynthesis of caffeine was closely related with the expression levels of *TCS1* [20]. We found that the *TCS1* was barely expressed in the ‘MYC’ group (Figure 6A). Therefore, the expression deficiency of *TCS1* results in lower accumulation of caffeine in the ‘MYC’ group. Furthermore, multiple MYB binding sites were found in the promoter of *TCS1* (Figure 6B). In tea plants, *CsMYB184* (TEA029017/CSS0026328) were proved to positively regulate caffeine biosynthesis [22]. However, the irrelevance was found between the expression level of *CsMYB184* and the accumulation of caffeine in this study (Appendix A). In this study, most of MYB genes were found preferentially expressed in ‘MYC’ groups with the fewest caffeine content (Figure 6C), indicating some of which potentially served as negative factors for biosynthesis of caffeine in tea plants. In fact, MYB transcription factor as a negative regulator have been well investigated, for example An R2R3-MYB regulated the flavonoid biosynthesis in plants [39]. However, further investigation was needed to perform to confirm which MYB genes perform such functions for negative biosynthesis of caffeine in tea plants.

## 5. Conclusions

In this study, the main secondary metabolites and the regulatory networks between Chinese varieties (‘BTZY’ and ‘BTSC’ group) and *Assam* varieties (‘MYC’ group) of tea plants were comparatively investigated by metabolomic and transcriptomic analyses. Our results suggested that, in green-leaf ‘MYC’ cultivars, most of the flavonoids were down-regulated, while the anthocyanin contents were not the lowest compared to purple-leaf ‘BTZY’ and green-leaf ‘BTSC’ group. In addition, transcriptomic analyses indicated that *SCPL1A* and *UGGT* potentially commonly contributed to the up-accumulation of the galloylated catechins content of Chinese varieties. Typically, the expression deficiency of *TCS1* was the key factor for remarkable down-accumulation of caffeine in ‘MYC’ cultivars. Moreover, the preferential expression of MYB family genes implied that they potentially served as a negative regulatory factor of caffeine biosynthesis by inhibiting the expression of *TCS1*. The present study uncovers the characteristics of flavonoids and caffeine accumulation and the key regulatory network in Chinese and *Assam* tea plants, which provide valuable information for to the selection and breeding of tea varieties.

## Figures and Tables

**Figure 1 genes-13-01994-f001:**
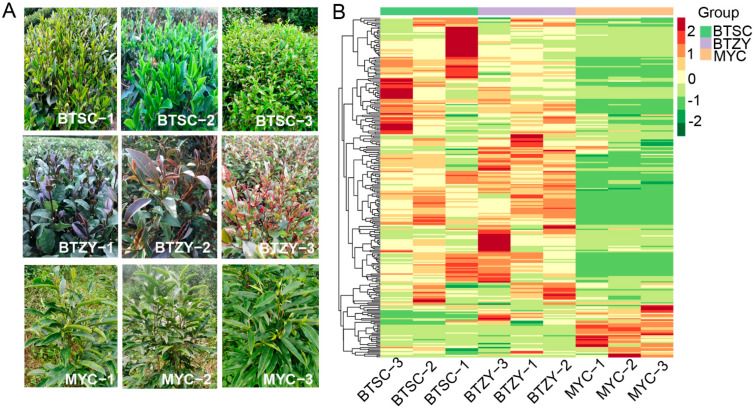
The morphological phenotypes and cluster analysis based on the metabolic profile. (**A**) the typical morphological phenotype of tea plants used in this study; (**B**) the cluster analysis of 259 metabolites based on the metabolome data.

**Figure 2 genes-13-01994-f002:**
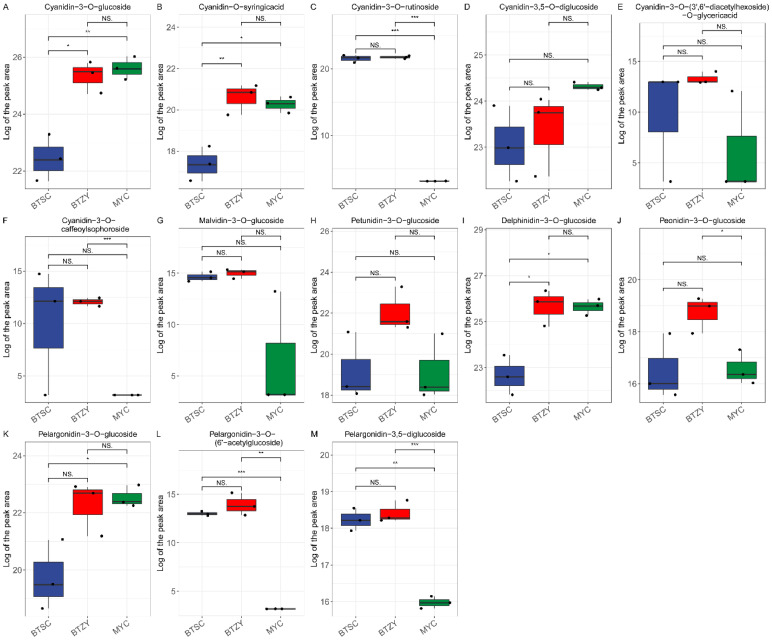
The differentially accumulated anthocyanins identified in this study. Differences between mean values of groups were compared using *t*-tests (“*”: *p* < 0.05; “**”: *p* < 0.01; “***”: *p* < 0.001; NS.: *p* > 0.05). (**A**–**H**) the value of logarithm (Log) of peak areas of different metabolites. The black circle refers to the value from various samples.

**Figure 3 genes-13-01994-f003:**
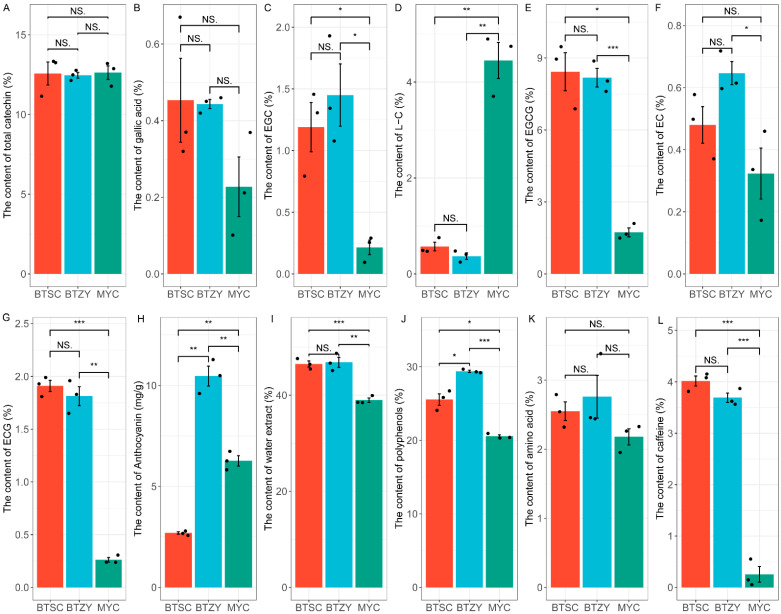
Measurement of bioactive components in three tea plants groups. Differences between mean values of groups were compared using *t*-tests (“*”: *p* < 0.05; “**”: *p* < 0.01; “***”: *p* < 0.001; NS.: *p* > 0.05). The black circle refers to the value from various samples.

**Figure 4 genes-13-01994-f004:**
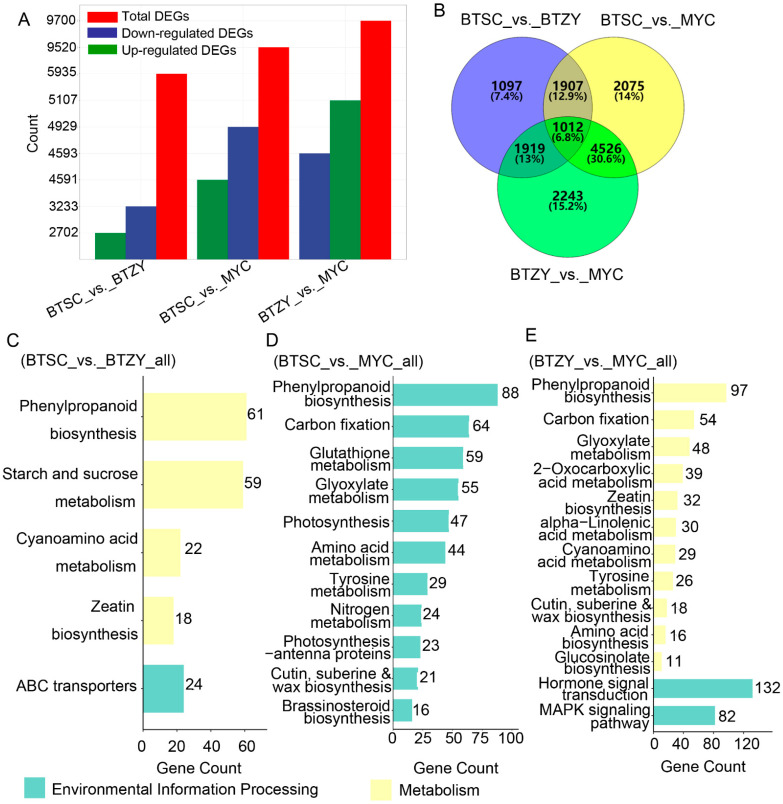
Identification of DEGs and KEGG enrichment in three groups. (**A**) Statistics of DEGs in three groups; (**B**) Venn diagram of DEGs in the three groups; (**C**–**E**) The main KEGG enrichment pathway of DEGs in three comparison groups.

**Figure 5 genes-13-01994-f005:**
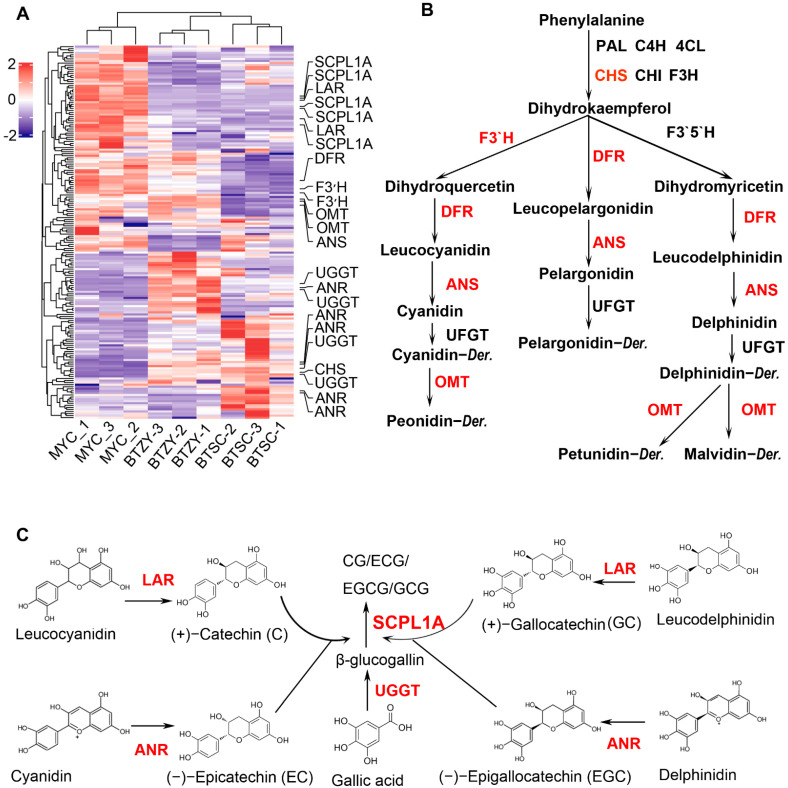
Identification of DEGs involved in anthocyanin and catechin biosynthesis. (**A**) the heat map of DEGs involved in anthocyanin and catechin biosynthesis; (**B**) the DEGs in anthocyanin biosynthesis pathway; (**C**) the DEGs in catechin biosynthesis pathway. The DEGs were noted by red font in (**B**,**C**).

**Figure 6 genes-13-01994-f006:**
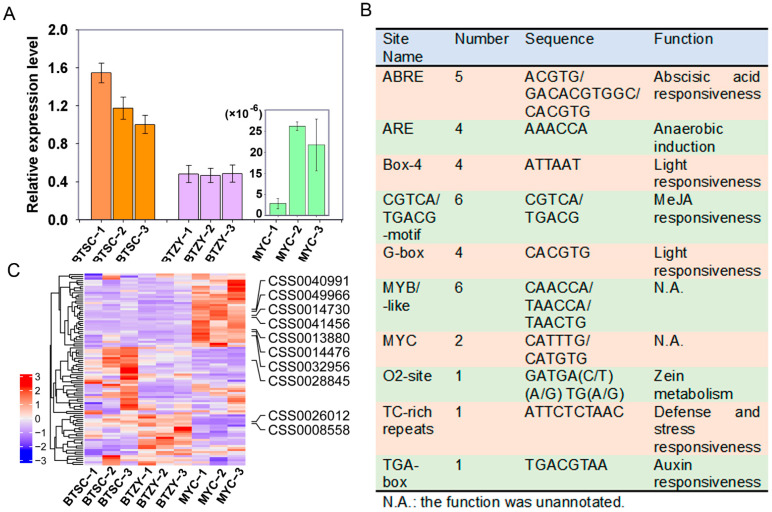
The expression pattern of *TCS1* and promoter elements analysis. (**A**)The relative expression level of *TCS1* in three sample groups; (**B**) Analyses of promoter elements of *TCS1*; (**C**) The expression pattern of MYB family genes.

## Data Availability

The data used in this study were deposited into the NCBI database and the accession number was: PRJNA883823.

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
