# Peer review of "Metabolomic and Transcriptomic Analyses Reveal the Characteristics of Tea Flavonoids and Caffeine Accumulation and Regulation between Chinese Varieties (Camellia sinensis var. sinensis) and Assam Varieties (C. sinensis var. assamica)"

_genes, 2022, doi:10.3390/genes13111994_

Round 1

Reviewer 1 Report

1. The introduction, towards the end, contain certain sections that should be included in materials and methods and results sections. These sections should be removed from introduction. Further, the objective of the research should be clearly stated at the end of the introduction. 

2. Certain abbreviated forms have been used without giving their expanded terminology at the beginning (example GO on page number 9). This should be corrected.

3. The conclusion should be improved by adding the potential applications or the usefulness of findings.

4. Some language editing is required to bring out the proper scientific meaning and to get rid of the grammatical mistakes.

Author Response

Dear reviewer,please check the attachment,thank you!

Reviewer 2 Report

The manuscript is a useful piece of work. Minor suggestions that need to be incorporated:

1. Avoid repeating sentences for example under plant material.

2. Check the grammar thoroughly.

3. Discussion needs to be strengthened in the light of existing literature about flavonoid and catechins in tea. 

Author Response

(The authors gave the same response as above.)
